# Pasteurized *Akkermansia muciniphila* Ameliorate the LPS-Induced Intestinal Barrier Dysfunction via Modulating AMPK and NF-κB through TLR2 in Caco-2 Cells

**DOI:** 10.3390/nu14040764

**Published:** 2022-02-11

**Authors:** Mengxuan Shi, Yunshuang Yue, Chen Ma, Li Dong, Fang Chen

**Affiliations:** 1National Engineering Research Center for Fruit and Vegetable Processing, Key Laboratory of Fruits and Vegetables Processing Ministry of Agriculture, Engineering Research Centre for Fruits and Vegetables Processing Ministry of Education, College of Food Science and Nutritional Engineering, China Agricultural University, Beijing 100083, China; mexush72@sina.com (M.S.); daisy_1022@126.com (Y.Y.); machen21@cau.edu.cn (C.M.); li_dong@cau.edu.cn (L.D.); 2Beijing DaBeiNong Biotechnology Co., Ltd., Beijing 100193, China

**Keywords:** *Akkermansia muciniphila*, intestinal barrier function, TLR2, AMPK, NF-κB, Caco-2

## Abstract

*Akkermansia muciniphila* is well known for the amelioration of inflammatory responses and restoration of intestinal barrier function. The beneficial effect of *A. muciniphila* occurred through contacting Toll-like receptor 2 (TLR2) on intestinal epithelial cells by wall components. In this case, the downstream mechanism of pasteurized *A. muciniphila* stimulating TLR2 for ameliorated intestinal barrier function is worth investigating. In this study, we evaluated the effect of live and pasteurized *A. muciniphila* on protecting the barrier dysfunction of Caco-2 intestinal epithelial cells induced by lipopolysaccharide (LPS). We discovered that both live and pasteurized *A. muciniphila* could attenuate an inflammatory response and improve intestinal barrier integrity in Caco-2 monolayers. We demonstrated that *A. muciniphila* enhances AMP-activated protein kinase (AMPK) activation and inhibits Nuclear Factor-Kappa B (NF-κB) activation through the stimulation of TLR2. Overall, we provided a specific mechanism for the probiotic effect of *A. muciniphila* on the intestinal barrier function of Caco-2 cells.

## 1. Introduction

*Akkermansia muciniphila* is an anaerobe and Gram-positive bacterium, representing approximately 1% to 3% of the total intestinal microbiota (10^9^ cfu/g) [1]. *A. muciniphila* generously distributes in the intestinal mucus layer and can contact epithelial cells at the villi tips more easily than other non-mucin-degrading bacteria [1,2]. *A. muciniphila* mainly uses intestinal mucins as its source of carbon and nitrogen and produces short-chain fatty acids, such as acetate and propionate [1]. Dietary supplementation enriches the abundance of *A. muciniphila* in mice, such as the dark sweet cherry [3], alpine bearberry, lingonberry, and cloudberry extracts [4], and inulin [5], fucoidan [6], polysaccharides from *Gastrodia elata* and *Ophiopogon japonicas* [7,8]. In addition, tryptophan and its derived metabolites could significantly promote the growth of *A. muciniphila* in vitro [9]. 

Recent studies indicate that supplementing with *A. muciniphila* can ameliorate metabolic syndrome, obesity, diabetes, and inflammatory bowel disease [10,11] in animals. *A.*
*muciniphila* improved the dextran sulfate sodium (DSS)-induced colitis by reducing the production of pro-inflammatory cytokines, such as TNF-α, IL-1β, IL-2, IL-6, and IFN-γ, and chemokines MCP-1 and MIP-1, and by enhancing the anti-inflammatory cytokine IL-10 [12,13,14]. In addition, *A. muciniphila* decreased the inflammatory response by reducing the infiltration of macrophages and the population of CD8^+^ cytotoxic T lymphocytes in mice [15]. Moreover, *A. muciniphila* reduced chronic low-grade inflammation by decreasing plasma lipopolysaccharide (LPS)-binding protein (LBP) and leptin and by inhibiting c-Jun N-terminal kinase (JNK) and Nuclear Factor-Kappa B (NF-κB) activation in metabolically abnormal mice [16]. Several studies reported *A. muciniphila* can enhance intestinal barrier integrity [17,18,19]. For instance, *A. muciniphila* improved the transepithelial electrical resistance (TER) of the epithelial monolayer and up-regulated the expression of zonula occludens-2 (ZO-2) in Caco-2 cells’ monolayer [20,21]. Moreover, researchers observed that *A. muciniphila* reduced gut permeability via increased expression of tight junction proteins including occludin, junctional adhesion molecule-3, claudin-3, -4, and -15 in mice with DSS-induced colitis [18,19,22]. 

Currently, several studies suggest that, similar to live *A. muciniphila*, inactivated *A. muciniphila* could ameliorate several diseases [15,18]. Supplementation with pasteurized *A. muciniphila* can significantly decrease LPS, insulinemia, and plasma cholesterol of obese human volunteers in a proof-of-concept study [23]. Moreover, oral administration of pasteurized *A. muciniphila* delayed the onset of colitis-associated colorectal cancer by modulation of CD8^+^ T cells in mice [15]. Animal experiments suggest *A. muciniphila* exerted its beneficial effects through contacting the intestinal epithelial cell of the host directly [22,24], and a special protein Amuc_1100 isolated from the outer membrane of *A. muciniphila*, was verified to bind human TLR2 directly in vitro [18,25,26]. However, there is little knowledge about how *A. muciniphila* improves the intestinal barrier through modulating TLR2. 

Research on the mechanism of *A. muciniphila*’s beneficial effects is deficient and warrants further study. AMP-activated protein kinase (AMPK) can act as an intracellular energy sensor for regulating energy homeostasis and metabolic stress and is activated by metformin and polyphenols [27,28,29]. Additionally, AMPK was found to promote the assembly of zonula occludens-1 (ZO-1) [30]. Therefore, we hypothesize *A. muciniphila* pretreatment may regulate tight junctions through the AMPK pathway in LPS-treated Caco-2 cells. We investigated the improving effect of live and pasteurized *A. muciniphila* on LPS-impaired barrier function in Caco-2 cell monolayers, respectively. We found that *A. muciniphila* may activate the AMPK and inhibit the NF-κB signaling pathway through modulating TLR2. This study provides a scientific basis for the understanding and application of *A. muciniphila* as probiotics. 

## 2. Materials and Methods

### 2.1. Growth Conditions and Heat-Inactivation of Akkermansia muciniphila 

*Akkermansia muciniphila* (DSM 22959) was purchased from the DSMZ (Braunschweig, Germany). The probiotic bacteria were cultured in Brain–Heart Infusion (Oxoid Ltd., Hampshire, United Kingdom) at 37 °C under strictly anaerobic conditions with H_2_/CO_2_/N_2_ (5/5/90, *v*/*v*/*v*) as a gas phase [18]. Cultures were centrifuged for 10 min at 16,000× *g* and the bacteria were washed two times with PBS. To obtain pasteurized *A. muciniphila* the part of live bacteria was inactivated by pasteurization for 30 min at 70 °C. The live or pasteurized *A. muciniphila* were resuspended in sterile DMEM medium under anaerobic conditions and the concentration was adjusted to 1 × 10^7^ CFU/mL to treat Caco-2 cells. 

### 2.2. Cell Culture 

The human Caco-2 cells line (1102HUM-NIFDC00087) were purchased from National Infrastructure of Cell Line Resource and cultured in Dulbecco’s modified Eagle’s medium (DMEM, 4.5 g/L glucose, Macgene, Beijing, China) supplemented with 10% (*v*/*v*) fetal bovine serum (FBS, Abwbio, AB-FBS0500, Shanghai, China) and 1% (*v*/*v*) Penicillin–Streptomycin (Gibco, Thermo Fisher Scientific, Rochester, NY, USA). The culture was kept at 37 °C in 95% air/5% CO_2_ and 95% humidity [31]. For expression of tight junction and inflammatory cytokine experiments, the Caco-2 cells were seeded at a density of 1 × 10^6^ cells/well into 6-well plates and incubated for 2 days at 37 °C.

The designated concentrations of 5-Aminoimidazole-4-carboxamide ribonucleotide (AICAR) (HY-13417, MedChemExpress, Monmouth Junction, NJ, USA) and dorsomorphin (HY-13418A, MedChemExpress) were used to investigate the cellular functions of AMPK [32], and C_16_H_15_NO_4_ (C29) (HY-100461, MedChemExpress) were used to inhibition the cellular functions of TLR2 [33]. 

### 2.3. Experiment Design

To investigate the protected effects of *A. muciniphila* on Caco-2 cells, an inoculum of 1 × 10^7^ CFU/mL live or pasteurized *A. muciniphila* was added to the cells before being treated with LPS. Caco-2/*A. muciniphila* co-cultures were incubated for 6 h at 37 °C with 95% air/5% CO_2_. After removal of the bacteria and the Caco-2 cells were washed 2–3 times with PBS. The 5 μg/mL LPS (L-2880, Sigma-Aldrich, St Louis, MO, USA) were added to the basal and apical side of the culture plates in the Transwell cell culture system, and cells were incubated for 6 h. After then, the Caco-2 cells were washed once with PBS and collected for further experiments. 

To build up of co-culture model of live or pasteurized *A. muciniphila* with Caco-2 cells, the cell viability was measured in different treatment groups. (A) The Caco-2 cells were co-cultured with 1 × 10^7^ CFU, 1 × 10^8^ CFU and 1 × 10^9^ CFU live *A. muciniphila* for 6 h (Appendix A). (B) The Caco-2 cells were co-cultured with 1 × 10^7^ CFU live *A. muciniphila* for 4, 6, and 8 h (Appendix A). (C) The Caco-2 cell was co-cultured with 1 × 10^7^ CFU live *A. muciniphila* for 4, 6, and 8 h after being stimulated by 5 μg/mL LPS for 6 h (Appendix A). Pasteurized *A. muciniphila* had no effect on the cell viability for each treatment (Appendix A). Therefore, for the next experiments, 1 × 10^7^ CFU live or pasteurized *A. muciniphila* was chosen to co-culture with Caco-2 cell for 6 h.

### 2.4. Measurement of TER

Intestinal barrier functions were evaluated by measuring the TER value in Caco-2 cell monolayers grown on polyester membranes in Transwell inserts [31]. The Caco-2 cells were seeded at a density of 5 × 10^5^ cells/well on inserts in the Transwell system (0.4 μm pore size costar, corning Incorporated, Corning, NY, USA) and placed into 12-well tissue culture plates. The medium was changed every 2 days until 21 days later when full polarization of the Caco-2 cell monolayer was achieved. The changes of TER were measured with the MILLICELL^®^-ERS voltohmmeter system (Millipore, Burlington, MA, USA).

Before starting each experiment, TER increased until 14–21 days when a steady state of higher than 450 Ω·cm^2^ was reached, indicating development of functional polarity and an intact monolayer, which is referred to as completed tight junction formation and maximized integrity of barrier function. To investigate the protective effect of *A. muciniphila* on intestinal barrier function, the TER value was measured before addition of *A. muciniphila* to the Caco-2 cell monolayers and after cells were incubated with *A. muciniphila* treatment for 6 h, and after induced of LPS for 6 h. Values were corrected for background resistance due to the membrane insert and calculated as Ω·cm^2^. The technical replicates for TER measurement were 3 times.

### 2.5. Monolayer Paracellular Permeability Determination 

The permeability of Caco-2 cell monolayer was determined by adding fluorescein isothiocyanate-dextran (FITC-dextran) as previously described with modifications [34]. The 100 μL 1 mg/mL FITC-dextran (20 kDa, Sigma-Aldrich, St Louis, MO, USA) was added to the apical side of the monolayer immediately after the last measurement of the TER value. Thereafter, the fluorescence intensity of the medium from each of the basolateral chambers was measured with excitation and emission wavelengths of 490 and 520 nm, respectively. The technical replicates for measurement were 3 times.

### 2.6. Transmission Electron Microscopy (TEM) 

The TEM were used to investigate the adhesion properties of *A. muciniphila* with Caco-2 cells. Approximately 1 × 10^6^ cells/well of Caco-2 cells were seeded in 6-well plates with or without cell slides. The Caco-2 cells were co-cultured with live or pasteurized *A. muciniphila* for 6 h at 37 °C and washed with PBS three times. The co-cultures without cell slides were fixed in 2.5% glutaraldehyde buffer solution and transferred to new 1.5 mL tubes for TEM [20]. The untreated Caco-2 cells were used as a control. Then, the samples were sent to Wuhan Servicebio Technology Co., Ltd. for further experiments. 

### 2.7. RNA Extraction and Quantitative Real-Time Polymerase Chain Reaction (Real-Time qPCR)

The real-time qPCR assay was performed as described previously [35]. At the end of co-culture, Caco-2 cells were homogenized with TRIzol reagent (Life Technologies, Carlsbad, CA, USA). Total RNA was isolated following the manufacturer’s instructions and was reverse transcribed using PrimeScript RT reagent Kit with gDNA Eraser (Takara, Kyoto, Japan). Quantitative real-time PCR was performed using SYBR Premix Ex Taq II Kits (Takara) in a Light Cycler 480 real-time PCR system (Roche, Basel, Switzerland). The reaction conditions included 40 cycles of three-stage PCR consisting of denaturation at 95 °C for 15 s, annealing at 60 °C for 1 min, and after an initial denaturation step at 95 °C for 10 min. The primers shown in Appendix A were used for PCR reactions. Expression levels of target genes were normalized to housekeeping gene GAPDH, and comparative quantification of gene expression was analyzed using the 2^−ΔΔCt^ method.

### 2.8. Cytokines Assay by Enzyme-linked Immunosorbent Assay (ELISA)

The secretion of cytokines in cell culture supernatants was determined using ELISA kits, according to the manufacturer’s protocols. Briefly, the supernatants of co-cultures were centrifuged 10 min at 5000× *g*. ELISA kits for TNF-α (MM-0122H1), IL-1α (MM-0056H1), IL-1β (MM-0181H1), IL-6 (MM-0049H1), IL-8 (MM-1558H1), TGF-β (MM-1774H1), and IL-10 (MM-0066H1) were obtained from Meimian Industrial Co., Ltd. (Yancheng, China).

### 2.9. Western Blot 

Western blot analyses were performed as described previously [32]. Total proteins were extracted from cultured Caco-2 cells using RIPA lysis buffer (Thermo Fisher, Waltham, MA, USA) and sonicated at 4 °C. The protein concentrations in cell lysates were quantitated by the BCA protein assay kit (Huaxingbio, Beijing, China) according to the manufacturer’s protocol. Forty-microgram protein was loaded and separated by SDS-PAGE (Mini-Protean 4–12%, Huaxingbio, Beijing, China) and then transferred electrically to PVDF membranes (Millipore, Bedford, MA, USA). The membranes were blocked using 5% bovine serum albumin (BSA) or 5% nonfat milk in Tris-buffered saline and Tween 20 (TBST, Huaxingbio, Beijing, China) for 1 h at room temperature. Then, the membranes were probed with primary antibodies overnight at 4 °C. The primary antibodies included occludin (33-1500, Invitrogen, Thermo Fisher, Carlsbad, CA, USA), 1:1000; claudin-1 (51-9000, Invitrogen, Thermo Fisher, Carlsbad, CA, USA), 1:1000; claudin-2 (51-6100, Cell Signaling Technology, Danfoss, MA, USA), 1:500; ZO-1 (61-7300, Invitrogen, Thermo Fisher, Carlsbad, CA, USA), 1:1000; TLR2 (#2229, Cell Signaling Technology, Danfoss, MA, USA), 1:1000; TLR4 (ab13556, abcam, Cambridge, MA, USA), 1:1000; phosphorylated AMPK (p-AMPK) (Thr172) (#2535, Cell Signaling Technology, Danfoss, MA, USA), 1:1000; AMPK (#2532, Cell Signaling Technology, Danfoss, MA, USA), 1:1000; phosphorylated NF-κB (p-NF-κB) (#3033, Cell Signaling Technology, Danfoss, MA, USA), 1:500; NF-κB (#8242, Cell Signaling Technology, Danfoss, MA, USA), 1:1000; CDX2 (ab76541, abcam, Cambridge, MA, USA), 1:1000; and β-actin (A1978, Sigma-Aldrich, St Louis, MO, USA), 1:1000. After three washes with TBST, the membranes were incubated with the anti-rabbit (Huaxingbio, Beijing, China) or anti-mouse secondary antibodies (Huaxingbio, Beijing, China) at 1:5000 dilutions for 1.5 h at room temperature. Bands were detected by using an Amersham Imager 600 (GE Healthcare Life Sciences, Pittsburgh, PA, USA). Densitometry analyses were performed by using Image J software (version 1.46r, National Institute of Health, Bethesda, MD, USA). The technical replicates for immunoblot analysis were 3 times. 

### 2.10. Statistical Analysis

All data were expressed as the mean ± S.D. Data of each test are consistent with normal distribution, statistical differences among different experimental groups were analyzed using one-way analysis of variance (ANOVA) with Tukey’s multiple comparisons test. All analyses were performed using SPSS (version 21, IBM Japan Inc., Tokyo, Japan) and GraphPad Prism version 6.0 (San Diego, CA, USA). Values of *p* < 0.05 were considered to be statistically significant, *p* < 0.01 were considered to be statistically extremely significant.

## 3. Results

### 3.1. Akkermansia muciniphila Ameliorated the LPS-Induced Inflammation in Caco-2 Cells

LPS treatment could significantly increase the inflammation level of Caco-2 cells. Fortunately, the pretreatment with live or pasteurized *A. muciniphila* significantly decreased the mRNA expression and secretion of TNF-α, IL-1α, IL-1β, IL-6, and IL-8 (*p* < 0.01, *p* < 0.05, Figure 1B,C). Moreover, TGF-β secretion levels increased in cells treated with live or pasteurized *A. muciniphila* (*p* < 0.05), and mRNA levels increased only by pasteurized *A. muciniphila* (*p* < 0.05, Figure 1D,E). In addition, the mRNA expression and secretion of IL-10 increased with live *A. muciniphila,* whereas pasteurized *A. muciniphila* only increased the mRNA level (*p* < 0.01, Figure 1D,E). Correspondingly, we found that pretreatment with *A. muciniphila* can reverse the LPS-induced up-regulation in p-NF-κB level (Figure 1F,G), and there is a significant difference in the p-NF-κB level between the PAL and LPS groups (*p* < 0.05). Therefore, we suggest that the *A. muciniphila* showed a pre-protect effect on the inflammatory status in Caco-2 cells induced by LPS.

### 3.2. Akkermansia muciniphila Ameliorated the Increased Barrier Permeability of Caco-2 Cell Monolayer Induced by LPS

The barrier functions were evaluated by measuring the TER value and FITC-dextran in Caco-2 cell monolayers. Results showed that LPS treatment (*p* < 0.05, Figure 2A,B) reduced the TER value and raised FITC-dextran. However, both pretreatment with live and pasteurized *A. muciniphila* could recover the LPS-induced increase in barrier permeation in Caco-2 monolayers (*p* < 0.01, Figure 2A,B). 

### 3.3. Akkermansia muciniphila Improved the Expression of ZO-1 and Decreased the Expression of Claudin-2 

The expression level of tight junction proteins can be used to evaluate the barrier integrity of the Caco-2 cell monolayer in this study. On the one hand, the pretreatment with live and pasteurized *A. muciniphila* significantly up-regulated the LPS-induced decline of the mRNA expression of claudin1 (*p* < 0.01, *p* < 0.05), whereas the mRNA expression levels of ZO-1 and claudin4 were increased by pretreatment with pasteurized *A. muciniphila* (*p* < 0.01, *p* < 0.05, Figure 3A). Furthermore, we observed the LPS-induced down-regulation of ZO-1 protein expression was rapidly enhanced by pretreatment with live and pasteurized *A. muciniphila*, respectively (*p* < 0.01, Figure 3B,C). On the other hand, the pretreatment with *A. muciniphila* consistently down-regulated the LPS-induced increase in claudin2 mRNA and protein expression levels. This suggests *A. muciniphila* can maintain barrier integrity by modulating the expression levels of tight junction proteins.

### 3.4. Akkermansia muciniphila Regulated TLR2 and Phosphorylated AMPK Expression in LPS Caco-2 Cells 

TLRs are involved in the promotion of barrier function, and TLR2 especially plays an important role in the recognition of *A. muciniphila* to epithelial cells. We applied TEM to observe how epithelial cells were recognized with *A. muciniphila* in co-cultured Caco-2 cells and to determine the expression of TLRs. TEM results showed that *A. muciniphila* adheres to the membrane of Caco-2 cells treated with both live and pasteurized *A. muciniphila* (Figure 4A). Additionally, the LPS-induced decline of TLR2 mRNA expression was significantly recovered in cells treated with pasteurized *A. muciniphila* (*p* < 0.05, Figure 4B), but the levels of TLR1 and TLR4 mRNA in all treatments did not significantly change (*p* < 0.05, Figure 4B). Consistently, TLR2 protein level showed an up-regulation in LPS-damaged Caco-2 cells by both *A. muciniphila* pretreatment but TLR1 and TLR4 were not affected (*p* < 0.01, *p* < 0.05, Figure 4C,F). 

Moreover, pretreatment with live or pasteurized *A. muciniphila* significantly increased the level of phosphorylated AMPK (p-AMPK) (*p* < 0.01, Figure 4D,E). We further treated Caco-2 cells with AMPK-activator AICAR and inhibitor dorsomorphin (Figure 4G–I) to verify the regulation of *A. muciniphila* on the AMPK pathway. We found that AICAR treatment showed similar p-AMPK activation to *A. muciniphila* treatment (*p* < 0.05, Figure 4G,I), whereas the *A. muciniphila*-induced recovery of p-AMPK in the LPS group was diminished by dorsomorphin (*p* < 0.01, Figure 4H,I). 

Then, we treated Caco-2 cells with TLR2 inhibitor C29 (Figure 4J–L) to determine whether AMPK activation is dependent on TLR2. Results showed that the influence of *A. muciniphila* on p-AMPK was only observed in cells without C29 treatment (*p* < 0.01), suggesting *A. muciniphila* modulates phosphorylation of AMPK by regulating TLR2 expression.

These results indicated that TLR2 and the AMPK signaling pathway were regulated by pretreatment with *A. muciniphila* and that TLR2 participates in the activation of AMPK.

### 3.5. Akkermansia muciniphila Increased the Level of ZO-1 and Decreased Claudin2 through TLR/AMPK Signaling 

Because AMPK is known to facilitate tight junction assembly, we next determined the effect of *A. muciniphila* on tight junction assembly through the AMPK pathway in Caco-2 cells treated with AICAR or dorsomorphin. Pretreatment with *A. muciniphila* increased the level of ZO-1 similar to AICAR (*p* < 0.01) and the effect was diminished after being treated with dorsomorphin (*p* < 0.01, Figure 5A–C). The expression of claudin2 was decreased in all groups treated with dorsomorphin (*p* < 0.01, Figure 5B,C). This indicated that inhibition of AMPK significantly eliminated the ability of *A. muciniphila* to ameliorate the level of ZO-1 decrease and claudin2 increase in LPS-induced Caco-2 cells.

As TLR2 is demonstrated to mediate the activation of AMPK, we investigated the effect of TLR2 and AMPK on the assembly of ZO-1 and claudin2 using Caco-2 cells treated with C29. C29 notably reduced the protein expression of ZO-1 but this decrease was not affected by subsequent treatment with AICAR or *A. muciniphila* (*p* < 0.01, Figure 5D–F). The improvement of ZO-1 by *A. muciniphila* in LPS-induced Caco-2 cells was eliminated after treatment with C29 (*p* < 0.01, *p* < 0.05, Figure 5E,F). The reduction of increased claudin2 expression by treatment with AICAR and *A. muciniphila* in LPS-induced cells was eliminated after treatment with C29 (*p* < 0.01, *p* < 0.05, Figure 5D,F). Results suggested that stimulation of TLR2 is important for tight junction assembly through AMPK activation in cells treated with *A. muciniphila*.

Together, these results showed that the activation of the AMPK signaling pathway through TLR2 was associated with increasing the level of ZO-1 and decreasing claudin2 by pretreatment with live or pasteurized *A. muciniphila*. 

### 3.6. Akkermansia muciniphila Regulated Caudal Type Homeobox 2 (CDX2) Expression in LPS Caco-2 Cells 

CDX2 is important in the assembly of tight junction proteins [30]. To determine the effect of CDX2 on ZO-1 and claudin2 assembly, the expression of CDX2 was measured. The Western blot analysis revealed that the level of CDX2 decreased markedly in LPS-treated Caco-2 cells (*p* < 0.01, Figure 6A,B), but live or pasteurized *A. muciniphila* reversed the protein expression of CDX2 significantly to the level found in the NC group (*p* < 0.01, Figure 6A,B). Moreover, pretreatment with pasteurized *A. muciniphila* showed higher up-regulation of CDX2 compared with live *A. muciniphila* groups (*p* < 0.05, Figure 6A,B). The results suggest *A. muciniphila* can improve transcription factor CDX2 in tight junction proteins.

## 4. Discussion

Mounting evidence indicates that *A. muciniphila* is beneficial for maintaining intestinal barrier integrity and immunologic response [13,19,24]. Previous studies have revealed that *A. muciniphila* activates TLR2 in vivo and in vitro [18,25]. However, the modulating mechanism of *A. muciniphila* is still obscure. In this study, we demonstrated that both live and pasteurized *A. muciniphila* strains can prevent LPS-induced epithelial barrier dysfunction and inflammatory response in Caco-2 monolayers through activating AMPK and inhibiting NF-κB. These findings provide a theoretical basis for understanding the mechanisms and application of pasteurized *A. muciniphila* for disease pre-protection.

Studies demonstrate the anti-inflammatory effect of *A. muciniphila* in vivo and in vitro [13,14]. In our study, the mRNA expression and secretion of pro-inflammatory cytokines, such as TNF-α, IL-1α, IL-1β, IL-6, and IL-8 was decreased, and anti-inflammatory cytokines TGF-β and IL-10 were increased in cells pretreated with *A. muciniphila*. Moreover, pasteurized and live *A. muciniphila* showed a similar effect in preventing inflammation disorders implementation. Consistently, we found that pretreatment with *A. muciniphila* significantly inhibited the DSS-induced activation of NF-κB in triggering inflammatory reactions [36,37], which was dependent on TLR2 stimulation. It is reported that the modulatory effect of NF-κB activated in the intestinal barrier is regulated by the recognition of TLR2 or TLR4 [38,39]. Moreover, the TLR2 was activated with Amuc_1100 of *A. muciniphila* in the human TLR2 and Caco-2 cell lines, respectively [18,26]. Therefore, we determined the mRNA and protein expression of TLR2 and TLR4 in this study. Our results showed that *A. muciniphila* indeed contacts Caco-2 cells, and pretreatment with pasteurized *A. muciniphila* significantly increased TLR2 expression, while TLR4 expression did not significantly change. This indicated that the anti-inflammatory effects of *A. muciniphila* are related to the stimulation of TLR2, not TLR4. 

Previous studies demonstrated that pasteurized *A. muciniphila* improved the expression of tight junctions, such as ZO-2, occludin, and claudin4, through interaction with TLR2 [18,25]. It is known that the maintenance of barrier integrity and their functions are determined by tight junction proteins [25,40], whose assembly can be regulated by the AMPK pathway [41,42]. Further, we determined the effect of pretreatment with *A. muciniphila* on AMPK phosphorylation in Caco-2 cells treated with LPS. In the treatment with the TLR2-inhibitor, we observed that C29 inhibited the activation of AMPK, which was not alleviated by the AICAR treatment, thus indicating the activation of AMPK is dependent on the TLR2. Additionally, the AMPK activated by *A. muciniphila* also vanished after the C29 treatment. Therefore, the TLR2 plays a key role in activating the AMPK pathway in Caco-2 cells treated with *A. muciniphila*.

ZO-1 can link the transmembrane proteins with the actin cytoskeleton in the cell and recruitment of other tight junction proteins to maintain intestinal function [43]. On the contrary, claudin2 is a pore-forming protein correlated with epithelial leakiness [44]. The up-regulation of ZO-1 and the down-regulation of claudin2 were simultaneously mediated by AMPK activation in mice and epithelial cell models [41,45,46]. In this study, we found the pretreatment with *A. muciniphila* resulted in a similar trend with that of the AICAR pretreatment. Consistent with that finding, dorsomorphin treatment eliminated the above effect, which was also diminished by the C29 pretreatment, suggesting that *A. muciniphila* may stimulate TLR2 and activate the AMPK pathway to regulate the protein expression of ZO-1 and claudin2. Additionally, *A. muciniphila* pretreatment up-regulated the CDX2 protein expression, which was eliminated by dorsomorphin. This means AMPK may promote ZO-1 assembly via CDX2 [30]. Consequently, pretreatment with *A. muciniphila* reversed the barrier dysfunction of LPS-induced Caco-2 cell monolayers significantly, including an increase in TER value and a decrease in FITC-dextran permeation. 

## 5. Conclusions

The present study indicated that pretreatment with live and pasteurized *A. muciniphila* can attenuate intestinal barrier dysfunction in LPS-induced Caco-2 cells by restoring the inflammatory cytokines toward the unchallenged control and by facilitating the assembly of tight junctions. The protective effects of *A. muciniphila* on barrier function were proved to ameliorate inflammation disorders by inhibition of the NF-κB pathway and to increase tight junctions via CDX2 mediation by activation of the AMPK pathway, both dependent on TLR2. Our findings revealed a specific mechanism responsible for the protective effect of *A. muciniphila* on the intestinal barrier function of Caco-2 cells (Figure 7). However, further research is needed to clarify how *A. muciniphila* stimulated TLR2 and to explain the minor difference in effect on claudin1 mRNA expression between pasteurized and live *A. muciniphila*. 

## Figures and Tables

**Figure 1 nutrients-14-00764-f001:**
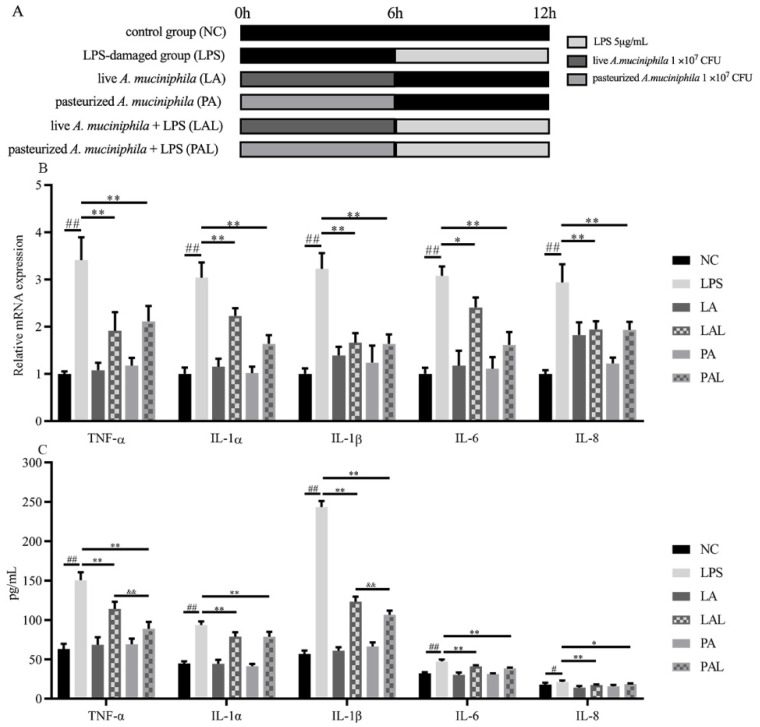
*A. muciniphila* inhibits LPS-induced inflammatory responses. (**A**) The experiment design of Caco-2 cells pretreatment with *A. muciniphila*. (**B**,**C**) The effect of *A. muciniphila* on expressions of TNF-α, IL-1α, IL-1β, IL-6, IL-8, (**D**,**E**) TGF-β, and IL-10 in Caco-2 cells. (**F**,**G**) *A. muciniphila* inhibits the activation of the NF-κB signaling pathway. ** *p* < 0.01, * *p* < 0.05, compared with LPS, ## *p* < 0.01, # *p* < 0.05, compared with NC group, && *p* < 0.01, LAL compared with PAL. Data are expressed as mean ± S.D. (*n* = 6). LPS, lipopolysaccharide. NF-κB, Nuclear Factor-Kappa B.

**Figure 2 nutrients-14-00764-f002:**
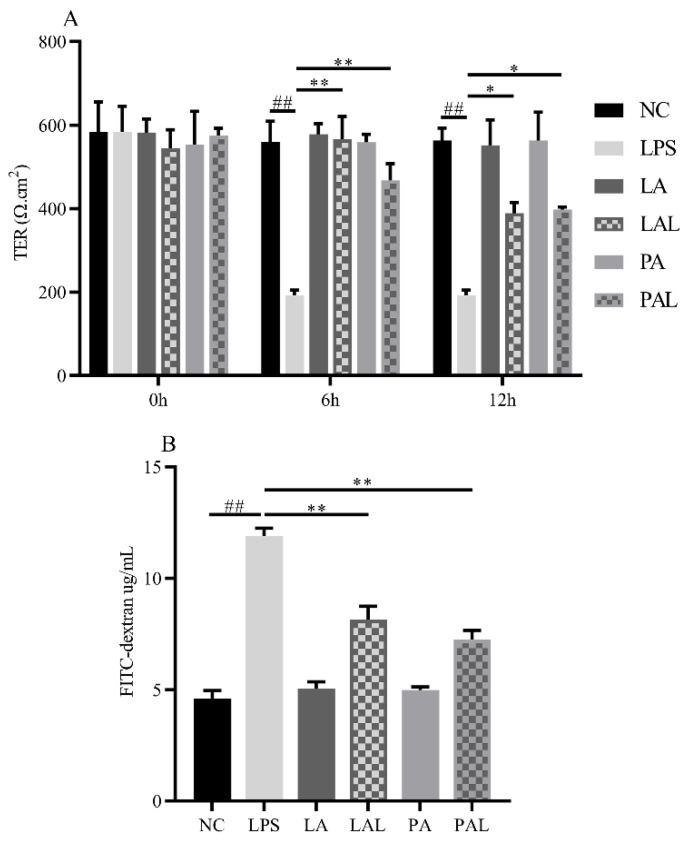
*A. muciniphila* improves the barrier function dysfunction induced by LPS. (**A**) The TER value and (**B**) FITC-dextran value of Caco-2 cell monolayers. ** *p* < 0.01, * *p* < 0.05, compared with LPS group. ## *p* < 0.01, compared with NC group. Data are expressed as mean ± S.D. (*n* = 6). TER, transepithelial electrical resistance. FITC-dextran, fluorescein isothiocyanate-dextran. NC, control group. LPS, lipopolysaccharide-damaged group. LA, live *A. muciniphila*. PA, pasteurized *A. muciniphila*. LAL, live *A. muciniphila* + LPS. PAL, pasteurized *A. muciniphila* + LPS.

**Figure 3 nutrients-14-00764-f003:**
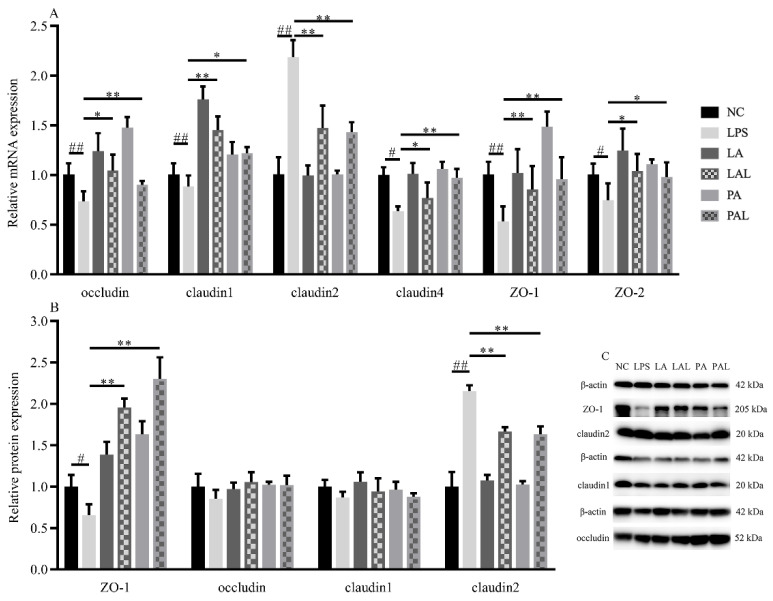
*A. muciniphila* affects the expression of tight junction proteins. (**A**) The level of relative mRNA expression, (**B**,**C**) and protein expression of tight junction proteins. ** *p* < 0.01, * *p* < 0.05, compared with LPS group. ## *p* < 0.01, # *p* < 0.05, compared with NC group. Data are expressed as mean ± S.D. (*n* = 6). NC, control group. LPS, lipopolysaccharide-damaged group. LA, live *A. muciniphila*. PA, pasteurized *A. muciniphila*. LAL, live *A. muciniphila* + LPS. PAL, pasteurized *A. muciniphila* + LPS.

**Figure 4 nutrients-14-00764-f004:**
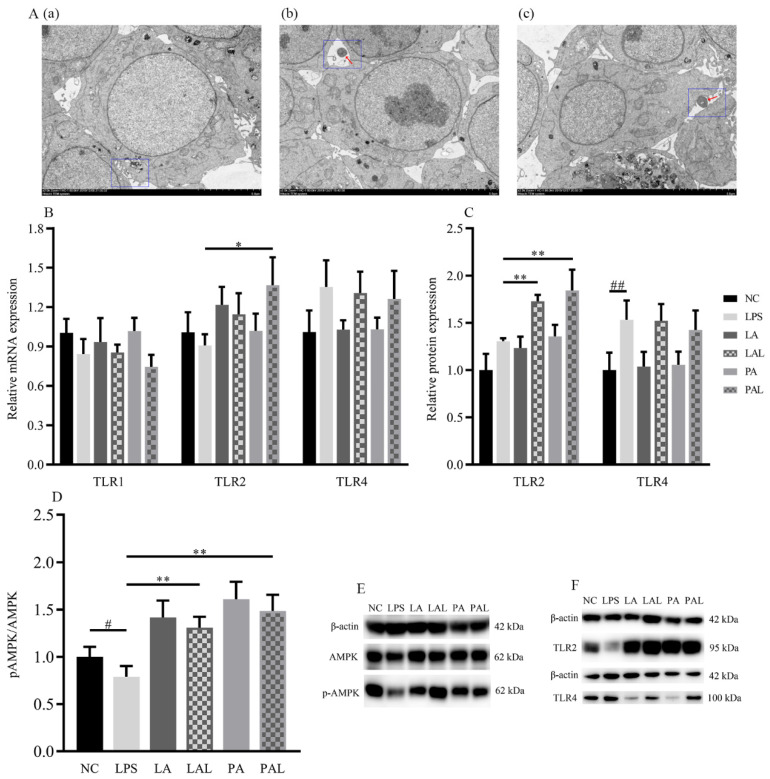
The activation of AMPK was mediated by *A. muciniphila* through TLR2 in LPS Caco-2 cells. (**A**(**a**–**c**)) The TEM images of *A. muciniphila* attaching to Caco-2 cells. The red arrows indicate *A. muciniphila*. The blue circles indicate cross-section between cells and *A. muciniphila*. Scale bars, 5 μm. (**B**,**C**,**F**) The expression of TLR2, TLR4, (**D**,**E**) and activation of AMPK in Caco-2 cells treated with *A. muciniphila*. (**G**–**I**) The activation of AMPK in Caco-2 cells treated with *A. muciniphila*, AICAR, or dorsomorphin. (**J**–**L**) The activation of AMPK in Caco-2 cells treated with *A. muciniphila*, C29, AICAR, or dorsomorphin. ** *p* < 0.01, * *p* < 0.05, compared with LPS group; ## *p* < 0.01, # *p* < 0.05, compared with NC group. Data are expressed as mean ± S.D. (*n* = 6). NC, control group. LPS, lipopolysaccharide-damaged group. LA, live *A. muciniphila*. PA, pasteurized *A. muciniphila*. LAL, live *A. muciniphila* + LPS. PAL, pasteurized *A. muciniphila* + LPS. AMPK, AMP-activated protein kinase. TLR, Toll-like receptor. AICAR, 5-Aminoimidazole-4-carboxamide ribonucleotide. C29, C_16_H_15_NO_4_.

**Figure 5 nutrients-14-00764-f005:**
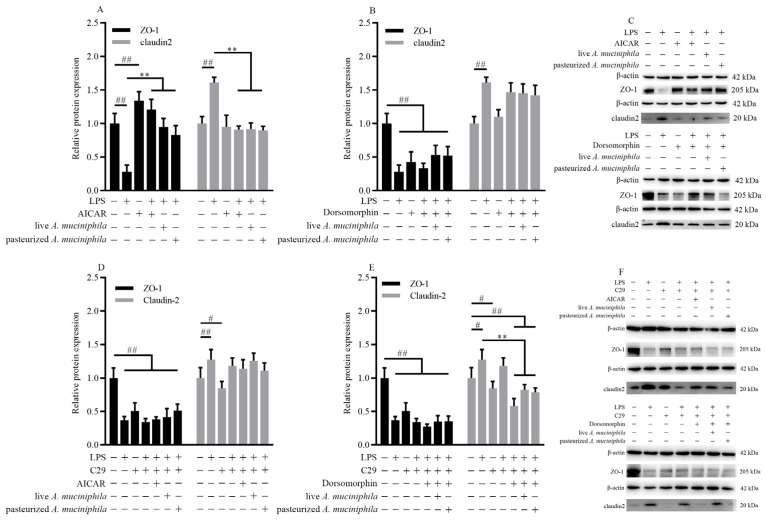
*A. muciniphila* mediated the expression of ZO-1 and claudin2 through activation of the AMPK signaling pathway dependent on TLR2 in Caco-2 cells. (**A**–**C**) The ZO-1 and claudin2 protein expressions were treated with AICAR or dorsomorphin in Caco-2 cells. (**D**–**F**) The level of ZO-1 and claudin2 treated with C29, AICAR, or dorsomorphin in Caco-2 cells induced by pretreatment with *A. muciniphila*. ** *p* < 0.01, compared with LPS group; ## *p* < 0.01, # *p* < 0.05, compared with NC group. Data are expressed as mean ± S.D. (*n* = 6). LPS, lipopolysaccharide. AICAR, 5-Aminoimidazole-4-carboxamide ribonucleotide. C29, C16H15NO4. ZO, zonula occludens.

**Figure 6 nutrients-14-00764-f006:**
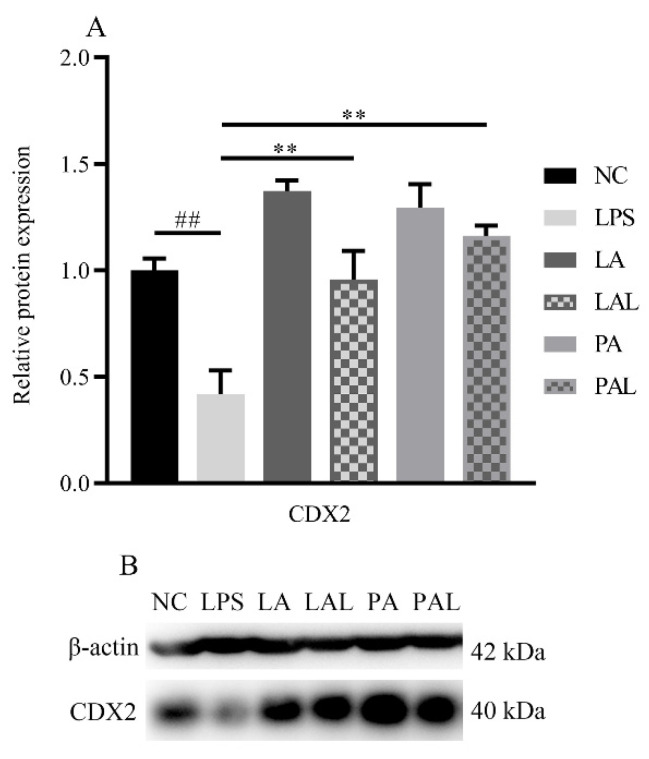
*A. muciniphila* mediated the expression of CDX2. (**A**,**B**) The protein expression of CDX2 in Caco-2 cells. ** *p* < 0.01, compared with LPS group. ## *p* < 0.01, compared with NC group. Data are expressed as mean ± S.D. (*n* = 6). NC, control group. LPS, lipopolysaccharide-damaged group. LA, live *A. muciniphila*. PA, pasteurized *A. muciniphila*. LAL, live *A. muciniphila* + LPS. PAL, pasteurized *A. muciniphila* + LPS. CDX2, Caudal type homeobox 2.

**Figure 7 nutrients-14-00764-f007:**
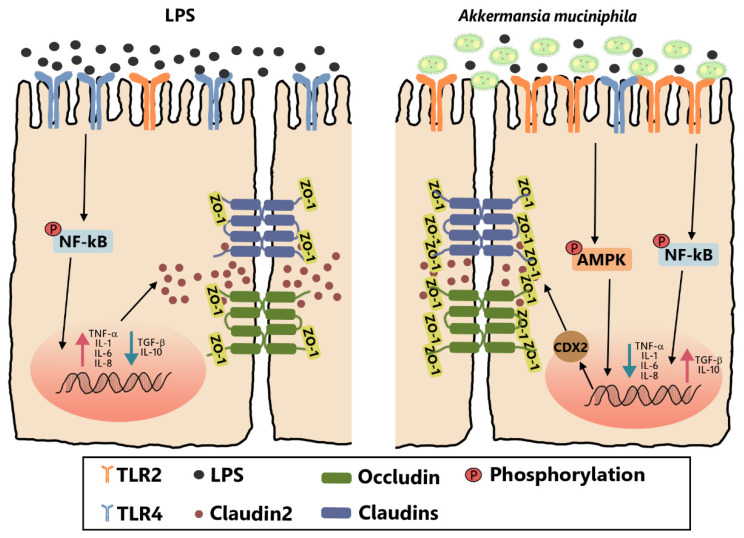
Pretreatment with live and pasteurized *A. muciniphila* can attenuate intestinal barrier dysfunction in LPS-induced Caco-2 cells by restoring the inflammatory response and by facilitating the assembly of tight junctions. The protective effects of *A. muciniphila* on barrier function were proved to ameliorate inflammation disorders by inhibition of the NF-κB pathway and to increase tight junctions via CDX2 mediation by activation of the AMPK pathway, both dependent on TLR2. TLR2, Toll-like receptor 2. LPS, lipopolysaccharide. ZO, zonula occludens. CDX2, Caudal type homeobox 2. AMPK, AMP-activated protein kinase. NF-κB, Nuclear Factor-Kappa B.

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
