# Peer review of "Pasteurized Akkermansia muciniphila Ameliorate the LPS-Induced Intestinal Barrier Dysfunction via Modulating AMPK and NF-κB through TLR2 in Caco-2 Cells"

_nutrients, 2022, doi:10.3390/nu14040764_

Round 1
Reviewer 1 Report
Contact time LPS 6H deposited in apical and basal. Why basal?
The authors say that Akkermansia muciniphila can reverse the pro-inflammatory reactions induced by LPS, but it is not a question of reversion but of non-implementation since it is a pre-treatment and not a treatment.
To say inversion it is necessary to observe a decrease after the stimulation with LPS
The increase in mRNA expression is not sufficient to confirm an increase in pro-inflammatory proteins. It is necessary to perform a cytokine assay or western blot of all these cytokines.
Concerning microscopy the authors announce an association with cell membranes. The authors do not show cross sections with brush borders and bacteria associated with them. These observations are needed to confirm the association.
The authors show an increase in TLR2 mRNA as well as an increase in TLR2 protein expression in the presence of pasteurized Akkermansia muciniphila.
However, a decrease in TLR4 expression is observed in the presence of Akkermansia muciniphila. This is not explained.
Conclusion:
he authors demonstrate experimentally in vivo the beneficial effect of live and pasteurized A muci on the dysfunctions set up by LPS. A muci seems to protect the intestinal cells, the first defense barrier of the intestine, from structural alterations by protecting the tight junctions. This protection seems to interact with TLR2.
Note to authors
The article is of good quality. The majority of the experiments have the correct controls. The choice of methods is appropriate. A cytokine assay is missing to validate the first results obtained. Be careful however, here the modalities are pretreatment and not treatment and this information is not always very clear.
Reviewer 2 Report
This article demonstrated that a mechanism for probiotec effect of A. muciniphila on the intestinal barrier function by using Caco-2 cells. The experimental design and the results seem to be exquisite and valid.
However, the following point would be considered to improve this article.
Minor point
1) Normally, it is reported that TLR2 signaling also leads to activation of NF-kappa B and increases the production of proinflammatory cytokines in immune cells.
On the other hand, TLR2 and AMPK passways lead to some inhibitory effects in Coca-2
cells demonstrated in this study. And, the inhibitory effect like this is also reported in another cells,
such as 3T3-L1.
Can we think that the TLR2 signaling, including the activation of AMPK and NF-kappa B lead to reduce the production of proinflammatory cytokines in non-immune cells, such as Coca-2 ?
Or is it due to the differences of TLR2 population and/or ligands as authors showed in Fig7 ?
Reviewer wants to read author’s idea about the differences of above reactions, if possible.
2) The expression style should be unified or the expression style is not appropriate.
Ex)
In “Discussion” section
Line332 : in vivo and in vitro(Italic) → in vivo(Italic) and in vitro(Italic)
In “References” section
Ref No.2 : Akkermansia m(there is a space)uciniphila → Akkermansia muciniphila
No.15 : Gut, 2020, 0:1-10 → Gut, 2020, 69, 1988-1997.
No.32 : Scientific Reports → Sci Rep
